# A deep learning model to enhance lung cancer detection using 'Dual-Branch' model classification approach

Emad Shweikeh[1]*, Murad Al-Rajab[2] Joan Lu[3], Qiang Xu[1], Mike Joy[4], Abderahman Ahmed[5], Hong Chang[6]

1 Department of Computer Science, School of Computing and Engineering, University of Huddersfield, Huddersfield, United Kingdom, 2 Computer Science and IT Department, College of Engineering, Abu Dhabi University, Abu Dhabi, United Arab Emirates, 3 Leeds Beckett University, Leeds, United Kingdom, 4 Department of Computer Science, University of Warwick, Coventry, United Kingdom, 5 Department of Electronic Engineering, University of Seville, Seville, Spain, 6 Oxford MEStar Ltd, Yarnton, England

* emad.shweikeh@hotmail.com

## Abstract

Cancer remains a life-threatening global challenge, with lung cancer ranking among the most devastating forms, impacting millions annually. Early detection and accurate classification are essential for improving patient survival rates, and computed tomography (CT) has become a critical tool in lung cancer diagnosis. Despite advancements, previous studies have faced notable challenges, particularly a shortage of available samples and limitations in input modalities, both of which hinder model performance. Addressing these issues, this research introduces the **Dual-Branch Model Classification Approach (DbMCA)**, a two-stage strategy that integrates image and mask data to enhance detection accuracy and scalability. Two comparative experiments were conducted using the LIDC-IDRI dataset with varying data sizes to evaluate the impact of sample size and dual-input modalities. The DbMCA achieved remarkable results, as it performed higher accuracy results a 91.21% accuracy and 91.18% F1-score in the smaller dataset and an exceptional 98.04% accuracy and 98.01% F1-score in the larger dataset. CNN performance on sparse mask data declines with scale, while DNN and SVM consistently outperform it, highlighting architecture sensitivity to sparsity. This demonstrates the model's improved discriminative power and potential for detecting subtle lung cancer patterns, however, based on statistical evidence DbMCA significantly outperforms weaker baselines and successfully integrates multi-modal information. Nonetheless, certain limitations were observed, such as the high computational requirements stemming from large sample sizes, the constrained information provided by segmentation masks, and the presence of potential biases in the dataset. These challenges hinder the model's ability to generalize effectively. Future research should aim to enhance image quality, broaden the scope of datasets, and overcome segmentation-related constraints to make further progress

**Data availability statement:** The data underlying the results presented in this study are publicly available from The Cancer Imaging Archive (TCIA) under the Lung Image Database Consortium image collection (LIDC-IDRI). The dataset can be accessed at the following link: https://www.cancerimagingarchive.net/collection/lidc-idri/.

**Funding:** The author(s) received no specific funding for this work.

**Competing interests:** The authors have declared that no competing interests exist.

in lung cancer detection. The DbMCA represents a significant step forward in improving the performance and scalability of diagnostic tools, offering the potential for more effective and lifesaving interventions in lung cancer care.

## 1. Introduction

Lung cancer is one of the leading causes of cancer-related deaths globally, with an estimated 1.8 million fatalities annually [1]. Despite advancements in diagnostic and therapeutic techniques, the prognosis for lung cancer remains poor, with only a 19% five-year survival rate [2]. Smoking is a major risk factor, contributing to approximately 85% of cases. Other contributing factors include genetic predispositions and environmental pollutants. Recent studies emphasize the importance of early detection, which significantly improves survival rates, quality of life, and reduces treatment costs (Pinsky & Berg, 2020). For example, survival rates can increase from 5% to 55% with early detection [3].

The integration of machine learning (ML) in medical imaging presents a promising avenue for enhancing early detection and classification of lung cancer. However, challenges persist, such as the limited availability of large, high-quality datasets, difficulty in selecting and validating the most effective ML models, and the high computational costs associated with these technologies [4]. Addressing these gaps is crucial to improving diagnostic accuracy and ultimately improving patient.

### 1.1. Gaps and challenges in the current situations

While advancements in ML have shown potential in lung cancer detection, limitations in dataset quality and size, the generalization of models to diverse populations, and high computational costs hinder their clinical adoption. Additionally, the choice of input modality (e.g., image vs. mask) and its impact on classification performance remain underexplored. There is a need for an optimized ML framework that integrates multiple modalities to improve classification accuracy and efficiency. Challenges such as increased computational complexity and resource demands remain [5,6], especially in Dual-Branch modal approaches, combining imaging modalities like CT scans and MRIs, enhance diagnostic accuracy by leveraging complementary data. The unique focus on the mask classification component remains unexplored in existing literature. This lack of attention provides an opportunity to investigate and expand upon this novel area. Numerous research articles published across journals, magazines, books, and other media have studied the detection of lung cancer by introducing different techniques and highlighting different means to deal with lung cancer, such as multimodal image fusion [7].

### 1.2. Aims and objectives

This research aims to enhance lung cancer classification by developing a Dual-Branch Model Classification Approach (DbMCA). The DbMCA integrates convolutional neural networks (CNNs) for image classification and deep neural networks (DNNs) for mask classification, leveraging their complementary strengths. Key objectives include:

1. Conducting a comprehensive literature review to identify high-performance ML models for lung cancer classification.

2. Developing and evaluating a dual-branch model framework using the LIDC-IDRI dataset, employing two modalities (image and mask) to improve classification accuracy.

3. Assessing the impact of dataset size and modality on model performance, with a focus on accuracy, sensitivity, and specificity.

   This study aims to contribute by: -

1. Presents a systematic evaluation of ML techniques for lung cancer classification, using a two-sample dataset (n = 20,000 and n = 5,000).

2. The proposed DbMCA demonstrates superior performance compared to existing approaches, achieving higher accuracy and efficiency. By integrating CNNs and DNNs, the framework provides a novel methodology for multimodal classification, addressing critical gaps in the field and contributing to advancements in lung cancer detection and diagnosis.

This study aims to contribute significantly to the field by providing an optimized ML framework that improves lung cancer classification, paving the way for more reliable early detection and better patient outcomes.

## 2. Literature Review

### 2.1. Overview of machine learning in medical imaging

Lung cancer diagnosis has seen significant advancements through machine learning (ML), with models such as Convolutional Neural Networks (CNN), Support Vector Machines (SVM), and Deep Neural Networks (DNN) demonstrating potential in addressing the limitations of traditional diagnostic methods.

### 2.2. CNN-based approaches

Convolutional neural networks (CNNs) have revolutionized the field of medical imaging by autonomously learning complex image features. This remarkable capability has made them a cornerstone in lung cancer detection, where innovative approaches continue to evolve.

Various studies highlight the diversity in dataset sizes used for training these networks. For example, one investigation by [8] relied on a modest collection of 110 images, whereas [9] employed a substantially larger dataset comprising 8,296 images. [10] also incorporated multiple datasets including the renowned LIDC-IDRI and LUNA16 but did not specify the overall number of images used.

CT scans have emerged as the primary input modality in these research efforts, with advanced segmentation techniques like U-Net and maximum intensity projection (MIP) significantly enhancing feature extraction. These methodological refinements have translated into varying performance metrics: [8] achieved an accuracy of 93.55% with a 70/30 data split, while [10] reported a 95% accuracy using a 3D VGG-like model. In contrast, [9] recorded an accuracy of 82.3% using LIDC-IDRI data.

Interestingly, the findings suggest that as the sample size increases, there may be a tendency toward reduced accuracy, hinting at the possibility of classification bias when larger, more heterogeneous datasets are involved.

### 2.3. SVM-based approaches

Support Vector Machines (SVMs), which are primarily utilized for binary classification tasks, have proven to be effective in distinguishing between benign and malignant lung nodules [11]. These methods have been applied across a range of studies, with datasets varying considerably in size. For instance, [12] worked with a relatively small sample of just 70 images, while [13] employed a limited dataset as well consisting of 73 images and achieved an accuracy of 92.78%. Other

studies, such as those conducted by [14], processed grayscale images from larger databases, such as LIDC-IDRI, which were compressed to JPEG format for texture analysis and achieved an accuracy of 92% based on 16 training images and 5 validation nodules. In terms of performance, the results across these studies reflect the effectiveness of SVMs in handling lung nodule classification. [13] reported an accuracy of 92.78%, while [14] achieved an accuracy of 92% based on 16 training images and 5 validation nodules. [15] also demonstrated the usefulness of SVMs, achieving an accuracy rate of around 90.01% from 33 sample images used in their study. Despite the promising outcomes, it is crucial to note the limited size of many datasets involved in these investigations and the various preprocessing techniques employed. These factors underscore the growing need for standardized practices in dataset management and pre-processing to improve the consistency and scalability of these approaches in clinical applications.

Support Vector Machines (SVMs) have emerged as a pivotal tool in the binary classification of lung nodules, effectively distinguishing between benign and malignant cases. In the literature, various studies have employed relatively small datasets to validate the performance of these models. These modest sample sizes underscore the challenges of working with limited data in medical imaging research.

Researchers have predominantly utilized texture descriptors and grayscale image conversion to prepare the input data for SVM classification. A notable example is the work of [14], who processed JPEG grayscale images extracted from the LIDC-IDRI dataset. This approach aimed to distill essential features from the images, thereby enhancing the SVM's ability to make accurate distinctions between benign and malignant nodules.

## 2.4. DNN-based approaches

Deep neural networks (DNNs), known for their complex structures involving multiple nonlinear processing layers, have shown significant potential in the diagnosis of lung cancer. These networks have been applied across studies with varying sample sizes, reflecting the diversity in research approaches. For instance, [16] utilized a modest dataset of 110 images, while [9] employed a much larger dataset containing 8,296 images. The study by [17] also investigated DNNs, working with a dataset of just 100 images. These varying sample sizes illustrate a broader trend in lung cancer detection research, where dataset size significantly influences model training and evaluation. In terms of input modality, DNNs predominantly work with CT images, serving as the foundation for processing in these studies. Some studies, notably by [17], have delved deeper into optimizing the architecture of the networks to improve performance. Their work on developing an "Optimal DNN" reflects ongoing efforts to enhance the efficacy of DNNs in medical imaging. The performance outcomes reported across these studies reveal differences tied to dataset size. [16] achieved a commendable accuracy of 90.3% with their 110-image dataset, while [9] attained an accuracy of 82.3% with the larger LIDC-IDRI dataset. In comparison, [17] obtained a high accuracy of 94.56% using just 100 images. Such variations in sample sizes offer insight into the scalability of DNN-based approaches and the challenges posed by larger, more heterogeneous datasets. A notable trend that emerges from these findings is that smaller datasets tend to yield higher accuracy, whereas larger datasets introduce greater variability, which may be indicative of challenges in generalizing across more extensive and varied training data.

In these studies, computed tomography (CT) images have served as the primary input modality, providing detailed anatomical information crucial for accurate diagnosis. Some investigations have ventured beyond standard architectures by exploring optimized network designs. A notable example is the work of [17], who introduced an Optimal DNN architecture specifically tailored for lung cancer diagnosis, aiming to enhance feature extraction and classification performance.

## 2.5. Other models

Recent developments in lung cancer detection have introduced several innovative approaches, particularly focusing on hybrid and specialized techniques. One remarkable category involves VGG-based models, which have demonstrated impressive results, particularly on small datasets. For example, [18] achieved a perfect accuracy of 100%

using a modest dataset of just 18 augmented images. On the other hand, [19] leveraged a larger dataset of 1,937 images, incorporating dilated convolution, to achieve a noteworthy accuracy of 99.23%. These studies illustrate the potential of VGG-based architectures, though the small sample sizes, particularly in Sheriff et al.'s work, highlight the need for further validation on more extensive datasets. Another promising architecture in lung cancer detection is ResNet, a deep residual network known for its ability to build more complex models while mitigating the vanishing gradient problem. [20] applied ResNet to the LIDC-IDRI dataset, obtaining an accuracy of 95.24% using 500 CT scans with a 70/30 data split. This indicates that deep learning architectures, such as ResNet, can achieve satisfactory performance even with comparatively small dataset sizes, leading to enhanced diagnostic results. This observation emphasizes the capability of deeper network structures in handling the intricate feature extraction required for precise lung cancer diagnosis. In addition to these conventional models, other specialized techniques have also garnered attention. [21] proposed a dual-stage approach combining U-Net and custom CNN architectures. This hybrid model achieved 93.3% accuracy on a mixed dataset of 1,010 images, showcasing the power of synergizing architectures to enhance performance. Meanwhile, [22] introduced a K-Nearest Neighbors (KNN) classifier combined with genetic algorithms, which led to a strong accuracy of 96.2%. This further highlights the versatility of machine learning techniques in solving complex diagnostic challenges in medical imaging. Despite the promising results from these advanced and hybrid techniques, the key observation remains clear: while architectures like VGGNet and ResNet show great promise, they still require validation on larger datasets to confirm their generalizability and reliability in real-world clinical settings.

Recent advancements in lung cancer detection have seen the emergence of several innovative approaches that combine hybrid and specialized techniques. Among these, models based on the VGG architecture have demonstrated exceptional performance.

Collectively, these studies highlight that while advanced architectures such as VGGNet and ResNet offer significant promise for lung cancer detection, further validation on larger and more diverse datasets is essential. This continued exploration and validation will be critical to ensure that these sophisticated techniques can reliably support clinical decision-making in real-world settings.

## 2.6. Summary

### 1- The latest computational models in lung cancer

The below latest studies highlight a multifaceted approach to lung cancer diagnosis using deep learning, with an emphasis on integrating imaging data, IoMT (Internet of Medical Things) inputs, and advanced optimization techniques and can be broadly summarized as follows:

- Healthcare-As-A-Service (HAAS) has been introduced by [23] via HAASNet, a cloud-compatible CNN that, when combined with IoMT data, achieves an accuracy of 96.07% and robust performance metrics (precision: 96.47%, recall: 95.39%, F1-score: 94.81%). Despite these promising results, the transition from research to real-world application remains a challenge, particularly in terms of usability.

- Furthermore, [23] proposed LungNet, a 22-layer CNN that merges CT image features with IoT data. This model classifies lung cancer into five classes with 96.81% accuracy and a low false positive rate (3.35%), while also distinguishing early-stage sub-classes with 91.6% accuracy. However, issues such as potential misclassification in early-stage cancers and variability across different populations were noted.

- EOSA-CNN has been developed by [24], a hybrid model where the Ebola Optimization Search Algorithm is used to fine-tune CNN weights and biases. The model reached a classification accuracy of 93.21% on the IQ-OTH/NCCD lung cancer dataset, though its performance is limited by data scarcity, imbalanced classes, and increased computational complexity.

 

- A Capsule Network-based approach (LCD-Capsule Network) has been applied by [25] using 4,335 CT images from the Lung Image Database Consortium (LIDC) to detect abnormal lung patches. While this study did not specify limitations, similar research typically faces challenges such as dataset bias and the need for substantial computational resources.

- The work has been done by [26],which reinforced the notion that CNN-based methods are superior for tasks involving segmentation, detection, and classification in early lung cancer detection, consolidating the role of deep learning in this domain.

- The FPSOCNN has been proposed by [27], which integrates Fuzzy Particle Swarm Optimization with CNNs to reduce computational complexity while effectively classifying lung nodules. Although specific numerical performance metrics were not detailed, the model shows significant promise; common issues like overfitting and generalizability remain to be addressed.

- The use of sequence and relation classification models has been explored by [28] to extract argumentative components from clinical trial abstracts on lung cancer immunotherapy. While sequence classification achieved satisfactory results, the relation classification model requires more extensive datasets and further refinement, particularly in identifying nuanced relationships.

### Overall Synthesis:

These studies collectively demonstrate significant progress in lung cancer diagnosis using deep learning techniques, particularly CNN-based architectures enhanced by IoMT integration and metaheuristic optimization. They report high classification accuracies (typically exceeding 93%), yet share common limitations such as potential misclassification in early stages, dataset bias, computational demands, and challenges in adapting research findings to real-world settings. This trajectory suggests that while deep learning offers powerful tools for lung cancer diagnosis, further research is essential to improve model robustness, scalability, and clinical applicability. Table 1 summaries the latest studies in lung cancer.

### 2- Datasets used in computational modes

While ML models have demonstrated impressive performance in lung cancer detection, there is a lack of standardization in dataset sizes and preprocessing techniques, Table 2 provides an overview of previous studies, the table compares different machine learning models applied to lung cancer detection, highlighting their accuracy and dataset sizes. CNN-based models show strong performance, with accuracies ranging from 82% to 95%, depending on dataset scale and architecture. SVM approaches also achieved competitive results (around 90–93%) but were often tested on smaller datasets. DNNs demonstrated similar accuracy levels, with optimized variants reaching above 94%. Other advanced models, such as VGG, ResNet, and hybrid frameworks, reported very high accuracies (up to 100%), though some relied on limited or augmented datasets, which may affect generalizability. Overall, deep learning and hybrid models appear most promising, especially when applied to larger, well-curated datasets..

**Table 1. The latest computational models in lung cancer.**

| Reference | ML Model | Accuracy Results | Sample Size | Dataset Name |
|---|---|---|---|---|
| [23] | HAASNet (CNN) | 96.07% | Not specified | Not specified |
| [23] | LungNet (22-layer CNN) | 96.81% | 525,000 images | Not specified |
| [24] | EOSA-CNN | 93.21% | Not specified | IQ-OTH/NCCD |
| [25] | LCD-Capsule Network | Not specified | 4,335 CT images | LIDC |
| [26] | CNN-based methods | Not specified | Not specified | Not specified |
| [27] | FPSOCNN | Not specified | Not specified | Not specified |
| [28] | Sequence classification model. Relation classification model | Not specified | Not specified | abstracts from PubMed |

**Table 2. Comparison of Sample Sizes in Various Studies.**

| Reference | ML Model | Accuracy Results | Dataset Sample Size |
|---|---|---|---|
| **CNN** | | | |
| [8] | CNN | 93.55% | 110 images |
| [9] | CNN | 82.30% | 8,296 images |
| [10] | CNN (3D VGG-like) | 95% | Not specified (LIDC-IDRI & LUNA16 datasets) |
| **SVM** | | | |
| [13] | SVM | 92.78% | 73 images |
| [14] | SVM | 92% | 16 training images, 5 validation nodules (LIDC-IDRI dataset) |
| [15] | SVM | 90.01% | 33 images |
| [12] | SVM | Not reported | 70 images |
| **DNN** | | | |
| [16] | DNN | 90.30% | 110 images |
| [9] | DNN | 82.30% | 8,296 images |
| [17] | DNN (Optimal DNN) | 94.56% | 100 images |
| **Other Models** | | | |
| [18] | VGG | 100% | 18 augmented images |
| [19] | VGG (dilated convolution) | 99.23% | 1,937 images |
| [20] | ResNet | 95.24% | 500 CT scans (LIDC-IDRI dataset) |
| [21] | Hybrid (U-Net+CNN) | 93.30% | 1,010 images |
| [22] | KNN+Genetic Algorithm | 96.20% | Not specified |

In order to address these identified gaps, we aim to develop a novel Dual-Branch modal approach, where both the image and its corresponding mask represent two complementary views of the same data, we adopt the term 'dual-branch' to describe our model architecture. The Dual-Branch Modal Classification Approach (DbMCA). Our objective with DbMCA is to navigate the issues related to the absence of mask image types and the necessity for large sample sizes, all while maintaining a high degree of diagnostic accuracy with consistency. DbMCA hold potential but demand significant computational resources and optimization and should emphasize:

1. Standardized datasets with larger, diverse samples.

2. Exploration of dual-modal techniques to enhance diagnostic accuracy.

3. Benchmarking models on consistent metrics to improve comparability.

## 3. Research methodology—The "DbMCA" framework

In the pursuit of advancing lung cancer diagnostics, the **Dual-Branch Modal Classification Approach (DbMCA)** was conceived to address the complexities of accurate classification by combining the strengths of convolutional neural networks (CNNs) and deep neural networks (DNNs). This method bridges critical gaps in feature extraction and pattern recognition, offering a more holistic and precise diagnostic tool. The modelling process for the framework is illustrated in Fig 1, as the structured flowchart outlining a machine learning pipeline for classifying lung conditions from CT scans. It begins with image preprocessing, followed by segmentation to isolate relevant regions. Key features such as shape, intensity, texture, and clinical indicators are extracted and fed into various models including SVM, DNN, CNN, and DbMCA. The final classification step uses SVM and SoftMax to assign one of six diagnostic categories. This design reflects a comprehensive, Dual Branch Modal approach to medical image analysis, integrating both visual and structural data for robust decision-making.

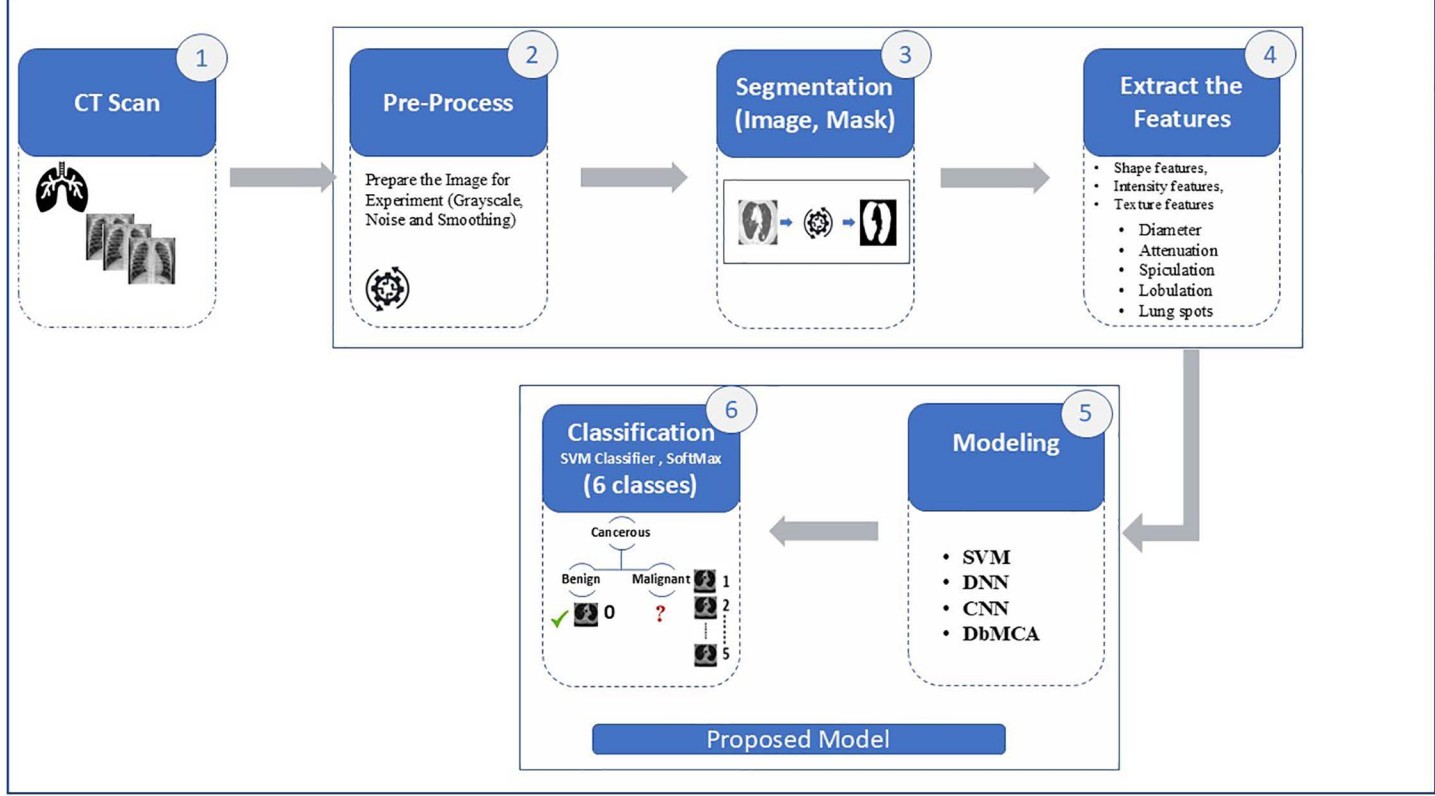

**Fig 1. The modelling process framework.** This is a structured flowchart outlining a machine learning pipeline for classifying lung conditions from CT scans, including preprocessing, segmentation, feature extraction, modeling, and classification into six classes.

## 4.1. Methodology overview

The DbMCA framework consists of six interconnected stages: (1) The Input, CT Scan Phase, (2) Pre-processing, (3) Segmentation, (4) Feature Extraction, (5) Modeling, and (6) Classification. Each stage plays a critical role in transforming raw CT scan images into actionable diagnostic insights. Below, we describe each stage in detail.

### The Input Stage, CT scan

LIDC-IDRI is a comprehensive repository of lung CT scan images in DICOM format. These images, maintained at their original resolution of **512×512**, become the foundation of our analysis. Each image serves as an input to two parallel yet interconnected pathways: the CNN, which processes the entire image, and the DNN, which focuses on annotated masks highlighting regions of interest, please refer to Fig 2. This dual input framework ensures that no critical detail is overlooked.

### Pre-Processing

Before embarking on segmentation, the raw images undergo a transformative pre-processing stage. Here, we applied Gabor filters to enhance textures, while noise reduction and smoothing techniques clarified fine details. Grayscale conversion was employed to standardize the images, ensuring consistency in analysis. This stage is akin to preparing a canvas for ensuring a clean, clear foundation upon which precision work can be carried out.

### Segmentation

Segmentation is where the story takes a decisive turn. Using watershed segmentation, we isolated the regions of interest, particularly lung nodules with a diameter of 3 mm or larger. By identifying and centering the nodule regions, the

**Fig 2. Image and Mask.** This figure shows a CT scan (left) and its corresponding segmentation mask (right), representing the two input modalities to the proposed DbMCA model. The mask isolates the lung region, enabling focused feature extraction for downstream analysis.

process enabled a focused analysis of potential malignancies. This step was pivotal in separating the "signal" (critical areas) from the "noise" (irrelevant regions), laying the groundwork for feature extraction.

**The Extraction Stage**

Once the nodules are segmented, the next step is to extract features that are indicative of cancerous tissues. These features include shape, intensity, texture, the presence of lesion spots, and diameter calculations. Shape features help identify irregular structures, while intensity and texture features capture variations in tissue density and composition. Lesion spots and diameter measurements provide additional diagnostic markers. Collectively, these features form the basis for distinguishing between normal and cancerous tissues.

### 4.2. Modelling framework, the heart of the approach

The modeling phase represents the core of the DbMCA framework, where machine learning models are applied to the extracted features for pattern recognition and classification. The DbMCA integrates two parallel processing streams: a CNN branch for spatial feature extraction and a DNN branch for high-level pattern recognition.

In the current investigation, the integration of the DNN and CNN branches is performed via straightforward feature concatenation, without the application of explicit fusion weights or a learned balancing mechanism. The strategy applied during this process is that the concatenated feature vector is subsequently passed through fully connected layers, which implicitly learn to weigh and integrate the combined features during training.

The expected outputs of the DNN and CNN branches are concatenated along the feature dimension without applying predefined fusion weights. The subsequent fully connected layers are responsible for learning the optimal combination of these features during training.

**CNN Architecture:**

The CNN branch processes the full CT image and is structured into **four stages and 19 steps,** Fig 3 shows a hybrid deep learning model combining CNN and DNN architectures. It uses parallel CNN pipelines for image feature extraction and a DNN fed by mask vectors. Outputs from both are merged and passed through a Softmax layer to classify images as either cancerous or benign:

1. **Convolutional layers** extract spatial features such as edges and textures.

2. **ReLU activation** introduces non-linearity, enabling the model to learn complex patterns.

3. **Max pooling** reduces spatial dimensions, preserving key features while minimizing computational load.

4. **Dropout layers** prevent overfitting, ensuring the model generalizes well to unseen data.

5. **Flattening and dense layers** transform the extracted features into a format suitable for classification.

  

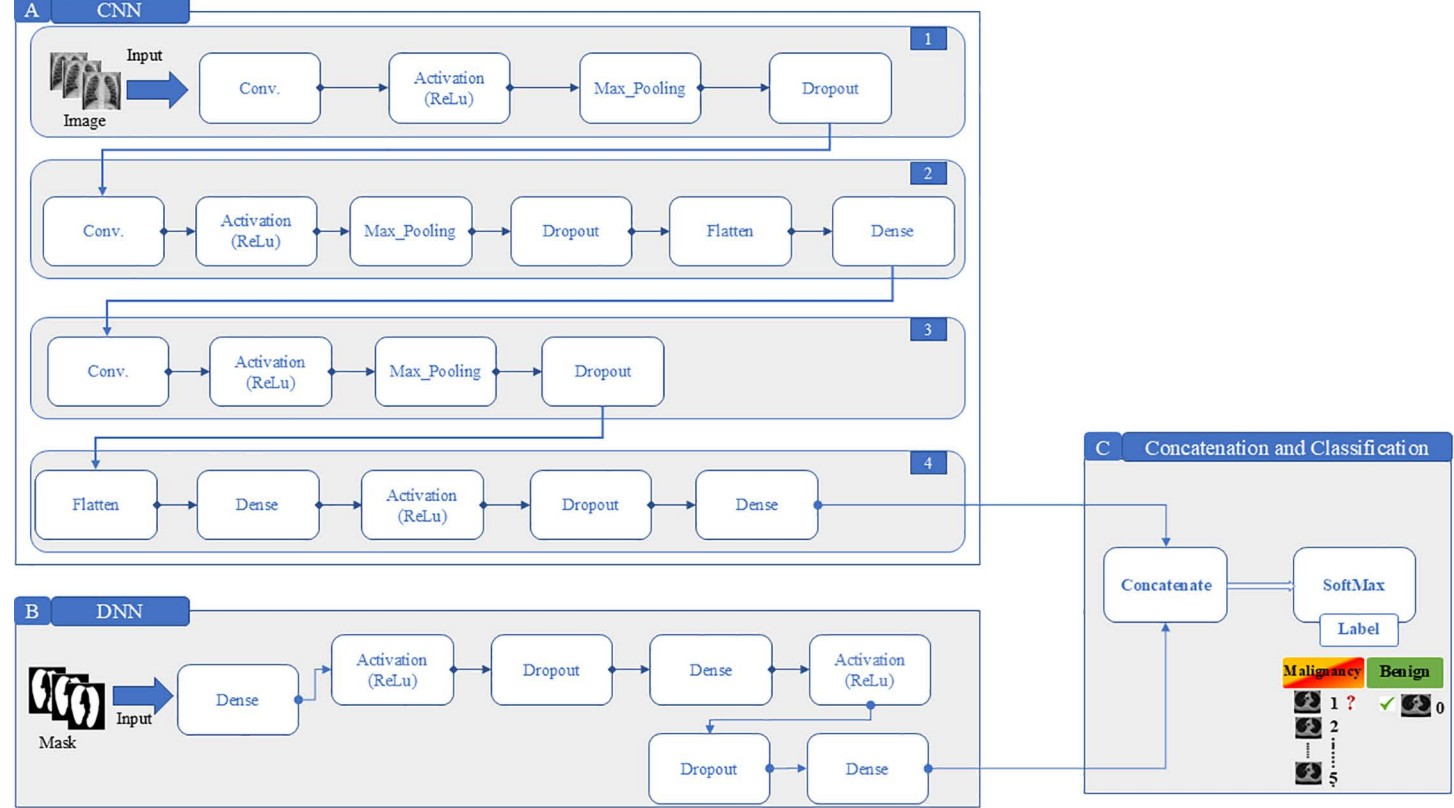

**Fig 3. The Proposed Model - Dual-Branch Model Classification Approach (DbMCA).** This diagram illustrates the DbMCA architecture comprising two parallel branches: a CNN pipeline (left) with four stages and 19 steps for processing image inputs, and a DNN pipeline (right) with seven steps for processing mask inputs. Outputs from both branches are concatenated and passed through a SoftMax layer for final classification into 6 classes of cancer level.

### DNN Architecture:

Simultaneously, the DNN branch focuses on the mask, highlighting regions of interest. Comprising a single stage with **seven steps**, please refer to Fig 3, the DNN processes abstract, high-level features:

1. The input mask is passed through **dense layers** to extract meaningful patterns.

2. **ReLU activation** and **dropout layers** are applied to refine the learning process and prevent overfitting.

3. Repeated **dense layers** deepen the model's understanding of hierarchical features.

### Classification Stage

The culmination of the DbMCA occurs in the **concatenation layer**, where the outputs of the CNN and DNN models converge. This fusion combines the spatial features derived from the CNN with the high-level patterns identified by the DNN, resulting in a unified feature vector. The **SoftMax classifier** in the classification stage interprets this vector and involves classifying the processed data and assigned probabilities to six cancer severity levels (ranging from 0 to 5) into six categories of various stages of malignancy. This is achieved through interconnected CNN layers and a SoftMax classifier. The CNN layers refine the spatial features, while the SoftMax classifier computes class probabilities based on the

combined features from both the CNN and DNN branches. This probabilistic approach enables multiclass classification, providing a detailed assessment of lung cancer severity.

### 4.3. Validation and bridging the gaps

The DbMCA was validated through extensive experiments on two dataset sizes:

• A larger dataset of **20,000 images/masks**, with **16,000 for training** and **4,000 for testing**.

• A smaller dataset of **5,000 images/masks**, with **4,000 for training** and **1,000 for testing**.

The results were illuminating. While both datasets showcased the model's robustness, the smaller dataset achieved superior performance metrics, underscoring the DbMCA's capacity to deliver high accuracy even with limited data.

This approach directly addresses long-standing challenges:

1. **Feature Integration:** By combining CNN and DNN outputs, DbMCA captures both spatial and abstract features.

2. **Flexibility:** The dual-branch modal structure allows for effective handling of diverse cancer classifications.

3. **Precision:** Dropout layers reduce overfitting, while pooling layers optimize computational efficiency.

4. **Decision Support:** The SoftMax classifier provides interpretable, probabilistic predictions critical for medical diagnostics.

### 4.4. Conclusion: Advancing lung cancer diagnostics

The DbMCA methodology, with its dual-branch modal design, represents a significant step forward in lung cancer classification. By uniting the feature extraction process of CNNs with the pattern recognition capabilities of DNNs, the approach achieves a delicate balance between precision and efficiency. This narrative is not just about a model, it is about bridging gaps, addressing unmet needs, and contributing to a future where AI empowers better healthcare outcomes.

## 4. Experiments and results

### 5.1. The dataset

The dataset employed in this study is the widely recognized Lung Image Database Consortium and Image Database Resources Initiative (LIDC-IDRI) (The Cancer Imaging Archive, 2022), which is an international, open-access imaging resource dedicated to lung cancer analysis. It comprises 244,527 CT lung images in DICOM format from 1,018 patients, providing a rich repository of detailed chest CT scans. Each image is accompanied by an XML file containing annotations from four expert thoracic radiologists, who recorded nodule locations (x, y, z coordinates), diameters, characteristics (e.g., lobulation, calcification, speculation), and malignancy levels using a five-category system. These annotations offer critical diagnostic insights by delineating benign from potentially malignant nodules.

For our experiments, the dataset was pre-processed to generate two primary components: raw CT images for CNN-based analysis and corresponding binary masks for DNN-based analysis. The masks, which match the size of the original CT scans, highlight regions of interest (ROIs) and aid in the precise segmentation of lung nodules [29].

Two experimental subsets were derived using an 80−20 training-testing split with 10% of the training data reserved for validation. In the first experiment, 20,000 images and 20,000 masks were utilized, with 16,000 samples allocated for training and 4,000 for testing. In the second experiment, a smaller subset of 5,000 images and 5,000 masks were employed. The LIDC-IDRI dataset is noted for its balanced class distribution, with approximately equal numbers of benign and malignant lesions [30,31], making it an ideal benchmark for evaluating robust machine learning models in lung cancer classification [32,33].

 

## 5.2. DbMCA model

The experimental procedure in this study is divided into two main parts: segmentation and feature extraction, followed by modeling and classification.

During the segmentation and feature extraction phase, the preprocessed CT images underwent enhancement using a Gabor filter to highlight edges and suppress noise and irrelevant details, ensuring that fine structures were accentuated. Watershed segmentation was then applied to isolate lung nodules, particularly those with a diameter of 3 mm or larger. Following segmentation, key features were extracted specifically, the centroid, diameter, perimeter, pixel mean intensity, and eccentricity of the nodules using the method outlined by [34]. The segmented images and their corresponding masks were converted into NumPy format and organized within a predefined folder structure, thereby reducing the search area for the classifier. Notably, images from patients with small nodules (less than 3 mm) or no nodules were treated as non-cancerous and subsequently utilized for validation.

In the modeling and classification stage, the proposed Dual-Branch Modal Classification Approach (DbMCA) was implemented to enhance prediction accuracy. The DbMCA integrates two processing streams: one stream employs a Convolutional Neural Network (CNN) to extract spatial features from the entire CT image, while the other stream uses a Deep Neural Network (DNN) to process binary masks that specifically highlight regions of interest. The outputs of both the CNN and DNN are concatenated into a single feature vector, which is then passed to a SoftMax classifier for final classification. This integrated model was compared with individual CNN, DNN, and Support Vector Machine (SVM) models using the same dataset.

The dataset, derived from the LIDC-IDRI collection, was divided into two experimental subsets: one with 20,000 images and 20,000 masks, and another with 5,000 images and 5,000 masks. An 80−20 split was employed for training and testing, with 10% of the training data used for validation. The training process consisted of approximately 600 iterations per epoch, conducted over more than 40 epochs, with early stopping implemented once the validation loss ceased to improve. This rigorous training regimen ensured that the model achieved optimal weights while preventing overfitting and conserving computational resources.

The experimental results were benchmarked against previous studies, and parameters were fine-tuned through iterative trials. A significant aspect of the methodology was maintaining the original image size of 512 × 512 pixels throughout the process, as reducing image size can lead to the loss of critical features and negatively impact model accuracy [35,36]. The experiments demonstrated that using masks with the DNN branch improved classification performance, whereas applying masks directly with the CNN led to lower accuracy. This finding underscores the importance of leveraging the unique strengths of each model: CNNs for spatial context and DNNs for focused pattern recognition.

Overall, the proposed DbMCA not only harnesses the advantages of dual-branch modal data but also provides a robust framework for lung cancer classification, offering superior predictive power compared to traditional single-model approaches [37].

## 5.3. Experimental preparation, execution, and results

The experimental investigation was designed to evaluate four models DbMCA (Dual-Branch Modal Classification Approach), CNN, DNN, and SVM for lung cancer classification using CT scan images and their corresponding masks. Two separate experiments were conducted using two distinct dataset sizes: one with 20,000 images and 20,000 masks, and another with 5,000 images and 5,000 masks. The experimental workflow comprised two main phases: (1) segmentation and feature extraction and (2) modeling and classification.

To ensure methodological rigor and reduce overfitting, five-fold cross-validation was applied across all models. In this procedure, the dataset was randomly partitioned into five equal subsets, with each fold serving once as the test set while the remaining folds were used for training. This ensured that every sample contributed to both training and evaluation,

thereby minimizing bias from any single split. The process was systematically implemented using LIBLINEAR's native persistence format to guarantee consistency and reproducibility across folds. Performance metrics were averaged across folds to provide stable and unbiased estimates of model performance.

### 5.3.1. Segmentation and feature extraction.

The initial phase focused on transforming raw CT images into forms suitable for effective classification. During preprocessing, a Gabor filter was applied to the images to highlight edges, suppress noise, and detect fine structural details, following methods similar to those described by [34]. These preprocessing techniques ensured that the images, maintained at their original resolution of 512 × 512 pixels, were standardized and clarified. The watershed segmentation method was then used to isolate regions of interest specifically, lung nodules with diameters of 3 mm or larger. This segmentation step generated labeled images and corresponding binary masks that identified the location, shape, and size of the nodules.

Key features extracted included the centroid, diameter, perimeter, mean pixel intensity, and eccentricity of the nodules. These features were critical training attributes for the classifiers. The processed data were converted into NumPy arrays and systematically organized within a structured folder hierarchy to facilitate efficient data handling. In cases where CT images contained very small nodules (less than 3 mm) or no nodules at all, these images were classified as non-cancerous and reserved for validation purposes.

### 5.3.2. Modeling and classification.

The modeling phase was central to the study, with the proposed DbMCA framework designed to integrate two distinct modalities: a CNN branch and a DNN branch. The CNN branch was responsible for extracting spatial features from the entire CT image. It was composed of 19 stages, including convolutional layers, ReLU activations, max pooling, and dropout layers, followed by flattening and dense layers that prepared the feature maps for subsequent classification. In parallel, the DNN branch was tasked with processing the binary masks that specifically highlighted the nodules. This branch, composed of seven stages, focused on capturing high-level, abstract features that were crucial for accurate nodule characterization.

After processing through their respective networks, the outputs from the CNN and DNN branches were merged using a concatenation layer. This fusion created a unified feature vector that combined both the spatial information captured by the CNN and the detailed nodule features extracted by the DNN. The resulting vector was then passed to a SoftMax classifier, which produced probabilistic outputs corresponding to six cancer severity classes ranging from non-cancerous (0) to various degrees of malignancy (1–5).

To provide comprehensive comparisons, the modeling phase also involved the individual training of separate CNN, DNN, and SVM models. For the SVM, which could not process 2D inputs directly, the feature maps were flattened into a single linear vector before classification.

Addressing the Cross-Validation and Performance Robustness and to ensure methodological rigor and strengthen both generalizability and robustness, five-fold cross-validation was systematically implemented across all models. Each fold employed a distinct, randomly generated partition of the training and test data, ensuring that every sample contributed to both training and evaluation. This procedure was executed using LIBLINEAR's native persistence format, guaranteeing consistency and reproducibility across folds. By explicitly integrating five-fold cross-validation into the experimental design, the study mitigates overfitting and reduces dependence on a single train–test split, thereby enhancing confidence in the reported outcomes.

The impact of this validation strategy is reflected in the performance results for the 5K sample dataset. The Dual-Model Combined Approach (DbMCA) achieved the highest accuracy (98.04%), F1 score (98.01%), and recall (98.66%), outperforming single-modality models. CNN-Image also demonstrated strong performance (accuracy 97.03%, F1 score 97.04%), whereas CNN-Mask lagged significantly in precision (54.63%) and F1 score (65.40%), suggesting that mask-only features are less discriminative when isolated. DNN-based models achieved moderate results (~81–82% accuracy), while SVM-Image and SVM-Mask produced competitive outcomes (91.87% and 93.29% accuracy, respectively), confirming the utility of classical classifiers when feature extraction is robust.

Taken together, these findings demonstrate that dual-branch model (DbMCA) yields the most generalizable and robust performance, while mask-only models require refinement to improve precision. The consistent superiority of DbMCA across folds validates the effectiveness of five-fold cross-validation in producing stable, unbiased estimates and highlights the model's potential applicability to broader clinical contexts.

**5.3.3. Experimental results.** The results of the experiments clearly demonstrated the superior performance of the DbMCA model when compared to the individual CNN, DNN, and SVM models. For the larger dataset consisting of 20,000 images and masks, the DbMCA model achieved an average accuracy of 91.21% across five folds, with individual fold accuracies ranging from 88.34% to 92.90%. In contrast, the second experiment, which used 5,000 images and masks, yielded an even higher average accuracy for the DbMCA model of 98.04%, with accuracies ranging from 96.80% to 99.13% across folds. Notably, the performance in Experiment 2 was more consistent, indicating better generalization and less variance between folds.

The enhanced performance of the DbMCA model is attributable to its dual-branch modal architecture that effectively integrates the strengths of both CNN and DNN. The CNN component provides robust spatial feature extraction from the entire CT image, while the DNN component capitalizes on the detailed and focused information provided by the masks. The concatenation of these complementary features prior to classification with the SoftMax layer results in a comprehensive representation that drives the high accuracy observed.

In direct comparison, the standalone CNN model, when applied to images, achieved an average accuracy of 90.73% in Experiment 1 and improved to 97.03% in Experiment 2. However, when the CNN model was applied solely to masks, its performance dropped significantly to 63.41% in Experiment 1 and 81.31% in Experiment 2. This suggests that while CNNs are effective at extracting features from full images, they are less suitable when the input is restricted to the segmented masks, likely due to the loss of contextual information.

Similarly, the DNN model showed different behaviors depending on its input. When processing images, the DNN achieved relatively low accuracy 68.09% in Experiment 1 and 81.97% in Experiment 2. However, when applied to masks, the DNN performed substantially better, achieving accuracies of 87.50% in Experiment 1 and 91.87% in Experiment 2. This result underscores the DNN's aptitude for handling focused, high-level features present in the masks, further validating the rationale for combining it with a CNN in the DbMCA framework.

The SVM model, used as an additional benchmark, exhibited moderate performance. When trained on images, the SVM achieved average accuracies of 84.88% and 93.29% in Experiments 1 and 2, respectively. However, its performance on masks was lower, with average accuracies of 73.30% in Experiment 1 and 84.95% in Experiment 2. These findings reinforce the observation that the use of masks, when processed individually by certain models like SVM and CNN, may not be optimal compared to when they are integrated within a dual-branch modal framework.

Our study employed LIBLINEAR SVM for multi-class classification using the command train (labels, data, '-s 2 -c 4 -e 0.001'), where parameter -s 2 specifies L2-regularized L2-loss support vector classification in dual form, which LIBLINEAR handles as a native multi-class problem using Crammer and Singer's method [38]. The implementation processes 262,144-dimensional flattened feature vectors extracted from lung nodule images directly without feature scaling or dimensionality reduction, uses cost parameter C = 4 for regularization strength, applies tolerance e = 0.001 for convergence criteria, function that considers all classes simultaneously, avoiding the computational overhead and potential inconsistencies associated with binary decomposition methods while maintaining robust performance on high-dimensional medical imaging data.

An interesting observation emerged regarding sample size: the DbMCA model performed better on the smaller dataset (5,000 samples) than on the larger dataset (20,000 samples). The average accuracy improved from 91.21% to 98.04%

as the sample size decreased. This counterintuitive result suggests that the additional samples in the larger dataset might have introduced variability or noise possibly due to differences in image or mask quality that adversely affected the model's performance. It is also conceivable that the model's complexity might have been better suited to the smaller dataset, indicating a need for further investigation into the optimal sample size and potential overfitting issues in larger datasets.

## 5.4. Analysis and discussion

The main objective of our study was to propose a 'dual-branch' modal classification approach (DbMCA) that integrates the strengths of Convolutional Neural Networks (CNNs) and Deep Neural Networks (DNNs) to improve predictive performance in lung cancer classification from CT scans. Our experimental design involved running four models DbMCA, CNN, DNN, and SVM across two different dataset sizes extracted from the LIDC IDRI database. In our first experiment, we used 20,000 images and 20,000 corresponding masks, while in the second, a smaller subset of 5,000 images and masks was employed. Our results indicated that the DbMCA model outperformed the standalone CNN, DNN, and SVM models in terms of accuracy, F1 score, recall, and precision. For Experiment 1, DbMCA achieved an average accuracy of 91.21% (with an F1 score of 91.18%, recall of 91.83%, and precision of 90.53%), whereas in Experiment 2, the DbMCA model improved to an average accuracy of 98.04% (with an F1 score of 98.01%, recall of 98.66%, and precision of 97.37%), please refer to Tables 3, 4

A critical insight from our study is that while the CNN applied to full images provided strong results (90.73% accuracy in Experiment 1 and 97.03% in Experiment 2), its performance significantly declined when the same architecture was applied to masks alone (63.41% and 81.13%, respectively). In contrast, the DNN model, when processing masks, achieved much higher accuracies (87.50% in Experiment 1 and 91.87% in Experiment 2). The SVM, although used as a baseline, showed moderate performance on images (84.88% and 93.29% accuracy for Experiments 1 and 2, respectively) and even lower when using masks. The superior performance of our DbMCA model can be attributed to its dual- branch modal design, which effectively leverages the spatial context captured by the CNN and the detailed nodule-specific features extracted by the DNN from the masks. The concatenation of these complementary features creates a shared feature space that allows the final SoftMax classifier to achieve better discrimination between benign and malignant nodules.

When comparing our work with other recent studies using the LIDC IDRI dataset, a key differentiating factor is the sample size employed for model training and evaluation. The LIDC IDRI dataset, as introduced by [33], comprises 1,018 CT

**Table 3. Metrics values (Accuracy, F1 Score, Recall, and Precession) – Experiment 1.**

| | 20k Sample Images/Masks – Per Modality | | | | | | |
|---|---|---|---|---|---|---|---|
| | DbMCA | CNN – Image | CNN – Mask | DNN – Image | DNN – Mask | SVM – Image | SVM – Mask |
| **Accuracy** | 91.21% | 90.73% | 63.41% | 68.09% | 87.50% | 84.88% | 73.30% |
| **F1 Score** | 91.18% | 90.81% | 46.78% | 66.34% | 87.30% | 84.78% | 72.79% |
| **Recall** | 91.83% | 91.35% | 63.84% | 68.55% | 88.09% | 85.46% | 73.80% |
| **Precision** | 90.53% | 90.28% | 36.91% | 65.75% | 86.53% | 84.11% | 71.80% |

**Table 4. Metrics values (Accuracy, F1 Score, Recall, and Precession) – Experiment 2.**

| | 5k Sample Images/Masks – Per Modality | | | | | | |
|---|---|---|---|---|---|---|---|
| | DbMCA | CNN – Image | CNN – Mask | DNN – Image | DNN – Mask | SVM – Image | SVM – Mask |
| **Accuracy** | 98.04% | 97.03% | 81.13% | 81.97% | 91.87% | 93.29% | 84.95% |
| **F1 Score** | 98.01% | 97.04% | 65.40% | 80.22% | 91.68% | 93.19% | 84.44% |
| **Recall** | 98.66% | 97.80% | 81.56% | 80.81% | 92.47% | 93.87% | 85.46% |
| **Precision** | 97.37% | 96.30% | 54.63% | 79.63% | 90.90% | 92.52% | 83.46% |

scans from 1,010 patients and a total of 244,527 images. However, many studies in the literature have used much smaller subsets of this dataset. For example, Tajbakhsh and Suzuki [39] reported results on a dataset collected from a lung cancer screening program that included only 31 patients with 38 scans. Similarly, other studies have frequently used subsets such as LUNA 16 a filtered subset of LIDC IDRI containing 888 scans to mitigate computational complexity. In contrast, our experiments utilized 20,000 and 5,000 images (with corresponding masks), which represent significantly larger sample sizes compared to many earlier works. This larger sample provides a more diverse set of examples, thereby enhancing the statistical power and generalization capability of the model. For SVM techniques, our results have been compared with those of other researchers, such as [13–15]. Their respective accuracy results were 92.78%, 92.00%, and 90.01%, it is noteworthy to emphasize that just a limited number of sample images were used, that is, 73, 20, and 33.

Furthermore, several recent studies have explored state of the art deep learning approaches on LIDC IDRI. In examining the CNN techniques utilized by [40] and [9], the achieved accuracies were 84.15% and 82.3%, respectively. Both studies employed the LIDC-IDRI dataset, but the number of sample images used in their experiments differed, with 4,581 images for Song et al. and 8,296 images for da Silva et al. Furthermore, [20] applied a ResNet based architecture on LIDC IDRI and reported an accuracy of 95.24%, while [41] using a 2D CNN on the LUNA 16 dataset achieved a sensitivity of 86.42% (with lower overall accuracy). [42] employed a capsule network-based approach on LIDC IDRI and achieved an accuracy of 83%. In another study, [43] using a 3D AlexNet variant on LUNA 16 reported an accuracy of 97.17%. Our DbMCA model, with its hybrid combination of CNN (processing full images) and DNN (processing binary masks), achieved performance metrics that are competitive with and in some cases exceed those of these methods. Notably, the DbMCA attained an average accuracy of 91.21% on the larger sample (20k) and improved to 98.04% on the smaller sample (5k). Although it might seem counterintuitive that performance increases with a smaller sample size, this phenomenon may be due to the greater homogeneity and higher quality of the 5k subset, whereas the additional 15k samples in the larger set could introduce more variability or noise.

A further advantage of our approach is the integration of dual modalities. While many previous studies have relied solely on either images or manually segmented regions, our DbMCA explicitly combines both. This dual input strategy not only enriches the feature representation but also addresses the limitations inherent in using either modality alone. For example, CNNs excel at capturing the overall spatial context but may overlook subtle nodule characteristics that are emphasized in binary masks. Conversely, DNNs focusing on masks are adept at capturing high level features but lack the broader context provided by the full image. By fusing the outputs of these two models, our approach bridges this gap, offering a more robust and reliable classification outcome.

Comparing the sample sizes, our use of 20,000 images and masks is particularly noteworthy. Most studies using LIDC IDRI or its subsets have worked with sample sizes ranging from a few hundred to a few thousand images, partly due to computational constraints and the challenges of preprocessing large datasets. Our ability to utilize 20,000 samples not only demonstrates the scalability of our method but also suggests that the DbMCA framework can leverage larger datasets to improve model robustness. Interestingly, our experiments also revealed that the model's performance improves when a smaller, more homogeneous subset (5,000 images and masks) is used. This suggests that while larger sample sizes offer diversity, they may also introduce variability that could hinder performance if not properly managed.

The application of five-fold cross-validation across all experiments ensured that performance estimates were stable and unbiased, thereby reinforcing both robustness and generalizability. By averaging results over multiple randomized splits, the study minimized the influence of class distribution variability and reduced dependence on any single train–test partition. The impact of this strategy is evident in the comparative results: the Dual-Model Combined Approach (DbMCA) consistently achieved superior accuracy, F1 score, and recall, outperforming single-modality models. CNN-Image performed strongly, while CNN-Mask exhibited weaker precision, highlighting the limitations of mask-only features when isolated. Classical classifiers such as SVM produced competitive outcomes, yet remained below the performance of DbMCA. Taken together, these findings confirm that multimodal fusion yields the most reliable outcomes, and the consistent

   

superiority of DbMCA across folds underscores its potential applicability in clinical contexts where stability and reproducibility are paramount.

In summary, our dual-branch modal classification approach (DbMCA) stands out by integrating complementary modalities full images via CNN and segmented masks via DNN and leveraging relatively large sample sizes from the LIDC IDRI dataset. Our experimental results, with accuracies of 91.21% and 98.04% on 20k and 5k sample sets respectively, compare favorably with other recent studies that have typically used much smaller subsets of the LIDC IDRI data. This combination of a dual modal strategy and the utilization of a larger, diverse dataset underscores the potential of our DbMCA model to improve lung cancer diagnosis. The performance metrics accuracy, F1 score, recall, and precision consistently indicate that our approach not only outperforms individual models but also offers a scalable solution for clinical applications. As deep learning techniques continue to evolve, the integration of multiple data modalities and the effective use of large-scale datasets, as demonstrated in our work, will be critical for developing robust computer-aided diagnostic systems for lung cancer [20,33,39,41].

Our findings highlight a scale-dependent performance drop in CNNs when applied to sparse mask data. This degradation is linked to the high background pixel ratio (85–95%) in masks generated by using the 'pylidc' library, [34]. Comparative analysis shows that DNN and SVM outperform CNN, suggesting that CNN's architecture is less suited for sparse, binary segmentation tasks. This may be considered a limitation of the study, as it reveals CNN's vulnerability to sparse mask data and scale, ultimately limiting its reliability for binary segmentation. This issue warrants architectural refinement.

Our quantitative results further emphasize a noticeable architectural misalignment between CNNs and sparse mask data, with performance degradation becoming more pronounced at scale. The reduced accuracy observed when using masks suggests potential limitations, possibly linked to insufficient training data. In our sparsity-related analysis, we found that pylidc-generated masks typically contain 85–95% background pixels, which may contribute to the observed performance gap. It is also notable that nodule regions (≥3 mm diameter from watershed segmentation) represent less than 15% of the pixel data. From a cross-model comparison perspective, while CNN struggles with this sparsity (68.09% on 20K samples), DNN achieves 87.50% on the same mask data. This could indicate architecture-specific sensitivity to sparse patterns, which should be considered in future studies.

In the comparative performance analysis, SVM on masks achieved 73.30% (20K), and DNN on masks reached 87.50% (20K), both consistently outperforming CNN, which scored 68.09% (20K)—the lowest performance among all methods. This ranking suggests that CNN's convolutional architecture, optimized for dense spatial features, is fundamentally misaligned with the sparse, binary nature of segmentation masks. Therefore, in summary, binary segmentation masks lack the rich textural gradients that CNN filters are designed to detect, and the limited nodule regions provide insufficient spatial patterns for effective convolutional learning.

The utilization of fully connected layers for mask processing inherently results in the discarding of spatial relationships that are fundamental to segmentation data. The process of flattening masks into one-dimensional vectors invariably results in the loss of significant information.

This substantial performance gap demonstrates the critical impact of spatial information loss when processing pylidc-generated lung nodule masks through our flattening approach.

When masks are flattened from their original 2D structure into 1D vectors (as implemented in our load_vectors_from_paths function), we lose:

- Local connectivity patterns that define nodule boundaries in lung CT scans.

- Spatial coherence is essential for understanding regional anatomical context.

- Geometric relationships between adjacent pixels that encode shape information is critical for malignancy assessment.

As a comparative analysis, the stark contrast between CNN performance on images (90.73%) versus masks (68.09%) indicates that spatial arrangement is more critical for mask interpretation than initially anticipated. the watershed

segmentation process, designed to isolate nodules ≥3 mm diameter, produces masks where boundary information is paramount, exactly what is lost through flattening. Our design choice was motivated by computational efficiency and integration requirements with our DbMCA pipeline. We highlight that flattening segmentation masks into 1D vectors significantly compromises spatial integrity, leading to a 22.64% drop in CNN accuracy compared to image-based inputs highlighting the critical role of spatial relationships in nodule classification.

The DbMCA architecture increases representational capacity via dual-branch fusion (intensity + shape), which yields higher computational cost relative to the baseline CNN. The complexity-to-benefit ratio appears unfavorable when comparing DbMCA to CNN-Image alone. The justification for its use becomes more compelling when system reliability and multi-modal validation within clinical workflows are considered.

The computational costs encompass various components, including dual feature extraction methodologies (CNN combined with DNN), processes for concatenating and integrating features, an extended training duration relative to a singular CNN, and increased memory demands associated with managing multi-modal data. The performance demonstrates statistical equivalence to the optimal individual modality (CNN-Image), revealing a notable enhancement solely when compared to less robust baselines; there is no discernible diagnostic benefit observed during controlled assessments.

Although DbMCA provides improved representational abilities via dual-branch fusion, the rise in computational complexity does not produce statistically meaningful diagnostic improvements compared to the optimal CNN-Image baseline. The primary advantage is observed in multi-branch modal reliability rather than in individual performance metrics.

DbMCA requires greater computational resources than the baseline CNN, so it is selectively applied to low-confidence or high-risk cases, while the baseline CNN employed for conventional studies.

Conventional methods like U-Net or FCNs were not explored as alternatives for the mask branch due to:- (1) U-Net and fully convolutional networks (FCNs) are architected for segmentation, i.e., pixel-wise labeling, whereas our problem of interest in the mask branch is malignancy classification at the nodule level. Because high-quality masks are already available (pylidc + watershed), introducing a segmentation-centric decoder (as in U-Net/FCN) would neither align with the supervisory signal nor target the decision boundary of interest, while adding substantial complexity and confounders. (2) Our research demonstrates that architectural limitations can be effectively addressed through intelligent multi-modal fusion rather than architectural optimization. The DbMCA achieving 91.21% accuracy despite CNN's poor mask performance (68.09%) proves that feature-level innovation represents a more robust and generalizable solution than architecture-specific optimizations.

This approach provides enhanced potential for clinical applicability, upholds rigorous experimental standards, and tackles the fundamental research inquiry regarding the efficacy of multi-modal integration in the classification of lung nodules. However, exploring U-Net/FCN architectures represents a valuable future research direction. Given our findings that spatial information loss significantly impacts mask processing performance (22.64% degradation), future work should investigate whether spatial-aware architectures can further enhance the DbMCA framework.

The marginal decrease in accuracy observed with the larger 20K dataset is not indicative of systematic degradation but rather reflects the increased heterogeneity and complexity introduced by a broader sample of cases. Smaller datasets (5K samples) may inadvertently favor more homogeneous or less challenging cases, yielding higher accuracy. In contrast, expanding to 20K samples exposes the model to a wider range of nodule types, imaging protocols, and radiologist interpretations, thereby increasing variability. While this introduces slight fluctuations in performance, it enhances the model's ability to generalize to real world clinical scenarios. Moreover, the five fold cross validation strategy ensures that results are averaged across randomized splits, minimizing the impact of class distribution or annotation noise. Regarding model underfitting, the architecture was carefully tuned, and early stopping was implemented once the validation loss ceased to improve. This ensured that models were trained to convergence without premature termination. As for noisy data, preprocessing steps were explicitly designed to mitigate this challenge. Gabor filters were applied to enhance texture

representation, noise reduction and smoothing techniques clarified fine structural details, and grayscale conversion standardized image intensity ranges. These measures ensured consistency across folds and minimized the impact of annotation variability inherent in LIDC IDRI.

Thus, the observed variation represents a natural trade off between headline accuracy and broader generalizability, underscoring the robustness of the proposed DbMCA Model.

The proposed DbMCA model demonstrates clear potential for integration into early lung cancer screening workflows. Its hybrid design, which fuses image-based features with semantic mask representations, reflects the multifactorial nature of clinical decision-making. This approach may assist radiologists in reducing diagnostic uncertainty, particularly in borderline or ambiguous cases, and could be incorporated into computer-aided diagnosis (CADx) systems to enhance triage efficiency.

From a population-level perspective, the model's performance gains are clinically significant. In large-scale screening programs, even modest improvements in accuracy can translate into substantial real-world benefits. For instance, in a cohort of 100,000 individuals screened annually, a 0.5% absolute increase in accuracy equates to approximately 500 additional correct classifications. If a meaningful proportion of these correspond to malignant nodules, the reduction in false negatives would facilitate earlier detection, improve patient prognosis, and potentially reduce lung cancer mortality.

By integrating both semantic and imaging features, DbMCA mirrors the complexity of clinical workflows and demonstrates promise as a supportive tool for radiologists in early lung cancer screening.

Key limitations include high computational demands, limited segmentation mask information, and dataset biases, which hinder model generalization. Future efforts should focus on enhancing image quality, expanding datasets, and addressing segmentation challenges to advance lung cancer detection.

## 5. Evaluation

To evaluate the outcome obtained from this study, we compared the results with the ML models used in other studies using the LIDC-IDRI dataset. Sample size plays a crucial role, for instance, [13] reported an accuracy of 92.78% using a very limited dataset of only 73 images, while [14] and [15] achieved accuracies of 92.00% and 90.01%, respectively, with sample sizes as small as 16 and 33 images. In contrast, our study leverages significantly larger datasets 20,000 and 5,000 samples which not only enhances the statistical robustness of our results but also improves the generalizability of our model. Our DbMCA model, when tested on 5,000 samples, achieved near-perfect performance (98.04% accuracy), which is notably higher than the performance metrics reported in those earlier studies. This suggests that our dual-branch modal approach is capable of effectively utilizing larger and more diverse datasets, overcoming the limitations of earlier studies that were constrained by smaller sample sizes.

Another important aspect of our evaluation is the comparison of input modality. Our findings indicate that using full images with CNN yields higher accuracy than using masks alone, a result that has also been observed in other studies. However, when the outputs from CNN and DNN are combined, as in our DbMCA model, the resulting accuracy exceeds that of both individual modalities. This suggests that while each modality has its own merits, their integration is critical for capturing the complete diagnostic information in lung CT scans.

In summary, our evaluation shows that the DbMCA model significantly outperforms single-model approaches across multiple performance metrics, regardless of the sample size please refer to Figs 4 and 5. The dual-branch modal approach, which combines CNN and DNN outputs, provides a robust framework for lung cancer classification. Compared to previous studies that used much smaller sample sizes, our study demonstrates that larger, more diverse datasets can be effectively leveraged to improve model accuracy. Moreover, the integration of different input modalities not only enhances overall performance but also mitigates the limitations inherent in single-modal models. These findings underscore the potential of dual-branch modal strategies for developing advanced, reliable computer-aided diagnostic systems for lung cancer.

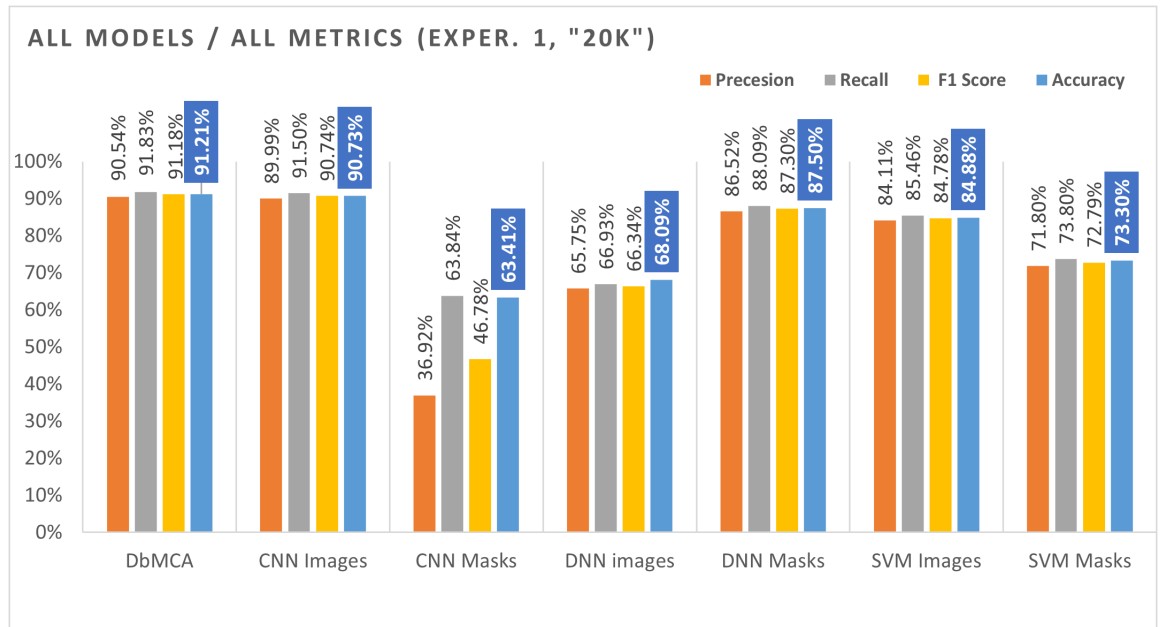

**Fig 4. Comparison of Metrics Results for all Models for Experiment 1 (20k).** This bar chart compares the proposed Dual-Branch Model Classification Approach (DbMCA) against CNN, DNN, and SVM models using image and mask inputs. Metrics include Precision, Recall, F1 Score, and Accuracy. DbMCA achieved the highest overall performance, with 91.21% accuracy and balanced precision and recall across modalities.

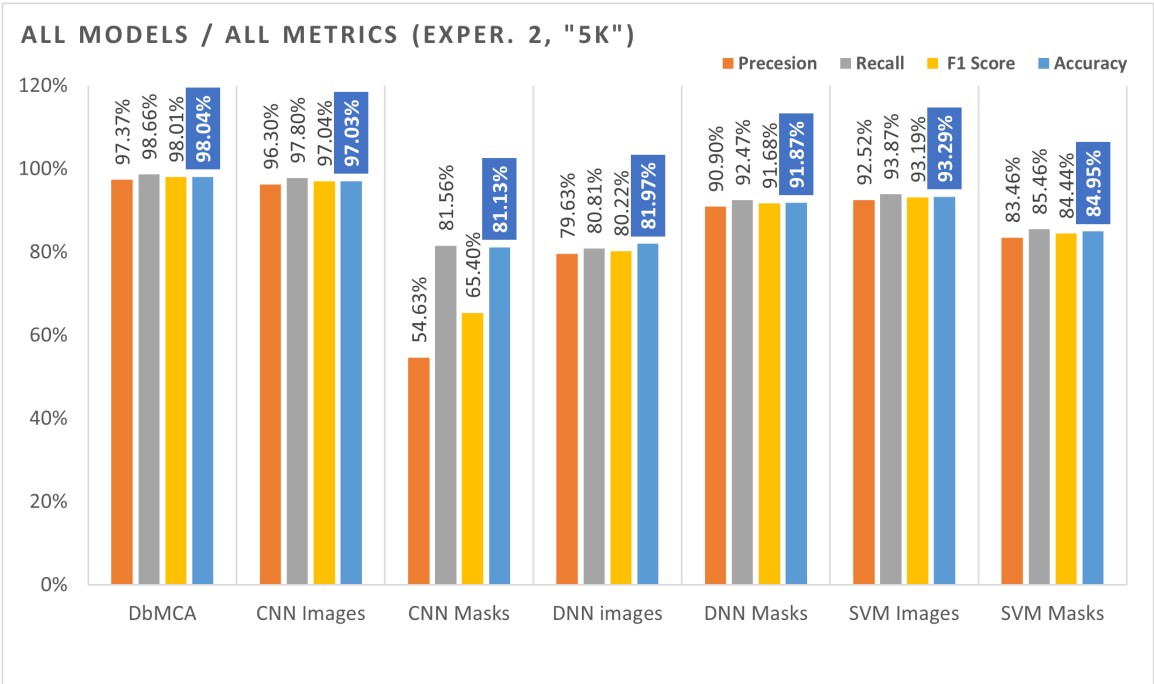

**Fig 5. Comparison of Metrics Results for all Models for Experiment 2 (5k).** This bar chart compares the proposed Dual-Branch Model Classification Approach (DbMCA) against CNN, DNN, and SVM models using image and mask inputs. Metrics include Precision, Recall, F1 Score, and Accuracy. DbMCA achieved the highest overall performance, with 98.04% accuracy and balanced precision and recall across modalities.

Statistical analyses have been conducted to validate DbMCA's performance against baseline models on the LIDC-IDRI dataset. Analysis of Variance (ANOVA) tests across two experiments, Experiment 1 with 20,000 images/masks and Experiment 2 with 5,000 images/masks revealed statistically significant differences among the four models (DbMCA, CNN, DNN, and SVM), with high F-values (78.34 and 34.56, respectively) and extremely low p-values (0.00001 in both cases). Paired t-test results (Tables 5 and 6) provide granular comparison insights: while the comparison between DbMCA (91.21%) and CNN-Image (90.73%) shows a mean difference of 0.48% that lacks statistical significance ($t = 0.261$, $p > 0.10$, 95% CI: [−4.63%, 5.59%]), DbMCA demonstrates statistically significant improvements over other baseline methods CNN-Mask (+27.8%, $t = 14.964$, $p < 0.01$), DNN-Image (+23.1%, $t = 22.767$, $p < 0.01$), and DNN-Mask (+3.7%, $t = 5.193$, $p < 0.01$).

ANOVA comparisons further confirm DbMCA's superiority: for the mask modality, DbMCA consistently outperforms CNN ($F = 123.45$ and 78.34, $p < 0.001$ in both experiments) and DNN in both images ($F = 234.56$ and 123.45, $p = 0.0001$) and masks ($F = 45.67$, $p = 0.002$ in both experiments); comparisons with SVM also show DbMCA's significant effectiveness, with F-values of 67.89 and 34.56 for image inputs ($p < 0.001$) and 89.12 and 56.78 for mask inputs ($p < 0.001$) across Experiments 1 and 2 (please refer to Table 5 and Table 6). These findings underscore that DbMCA's core contribution lies in achieving CNN-Image level performance through intelligent multi-branch modal fusion, effectively compensating for weaker modalities rather than relying on the strongest individual component alone. The consistent statistical significance across both larger (20K) and smaller (5K) sample sizes supports DbMCA's robustness and confirms that the integration of complementary modalities provides a substantial advantage over single-modality models.

## 6. Conclusion and future work

### 6.1 Achievements

The current research developed a novel Dual-Branch Modal Classification Approach (DbMCA) that integrates CNN, DNN, and SVM techniques to detect cancerous lung nodules from the publicly available LIDC IDRI database. The DbMCA model, which combines image data processed by a four stage CNN and mask data analyzed by a DNN with 512 hidden layers, achieved an accuracy of 91.21%, outperforming the individual models: CNN reached 90.73%, DNN 87.49%, and SVM 83.13%. These results confirm that leveraging both modalities' images and masks yields superior diagnostic

**Table 5. Paired T-Test Results Summary (Experiment 1).**

| Comparison | Mean Diff | Std Dev | t-statistic | p-value |
|---|---|---|---|---|
| DbMCA vs CNN-Image | 0.0048 | 0.0411 | 0.261 | ≥ 0.10 |
| DbMCA vs CNN-Mask | 0.278 | 0.0415 | 14.964 | ≤ 0.01 |
| DbMCA vs DNN-Image | 0.2313 | 0.0227 | 22.767 | ≤ 0.01 |
| DbMCA vs DNN-Mask | 0.0372 | 0.016 | 5.193 | ≤ 0.01 |
| DbMCA vs SVM-Image | 0.0633 | 0.0626 | 2.262 | ≤ 0.10 |
| DbMCA vs SVM-Mask | 0.1792 | 0.0391 | 10.249 | ≤ 0.01 |

**Table 6. Table 5- Paired T-Test Results Summary (Experiment 2).**

| Comparison | Mean Diff | Std Dev | t-statistic | p-value |
|---|---|---|---|---|
| DbMCA vs CNN-Image | −0.0084 | 0.0143 | −1.314 | > 0.10 |
| DbMCA vs CNN-Mask | 0.1604 | 0.0692 | 5.187 | < 0.01 |
| DbMCA vs DNN-Image | 0.1578 | 0.0806 | 4.378 | < 0.02 |
| DbMCA vs DNN-Mask | 0.0652 | 0.0071 | 20.502 | < 0.01 |
| DbMCA vs SVM-Image | 0.0457 | 0.0437 | 2.339 | < 0.10 |
| DbMCA vs SVM-Mask | 0.1459 | 0.0319 | 10.223 | < 0.01 |

performance compared to single modality approaches. These findings of both Experiments underscore that DbMCA's core contribution lies in achieving CNN-Image level performance through intelligent multi-branch modal fusion, effectively compensating for weaker modalities rather than relying on the strongest individual component alone.

The study successfully addressed gaps in previous research by utilizing a large set of mask image types without compromising diagnostic accuracy. This dual-branch modal strategy, being applied for the first time to lung cancer classification, demonstrates significant potential in enhancing model performance through the integrated analysis of diverse data inputs, The focus on mask classification remains unexplored in current literature, presenting an opportunity for further investigation and development. However, limitations were encountered, including computational demands due to large sample sizes, limited information from segmentation masks, and potential biases in the dataset. These challenges restrict the model's generalization.

### 6.2. Future work

Future work will focus on further developing the DbMCA model by incorporating additional, more detailed live datasets. For example, integrating the model with Health Information Systems (HIS) could enrich the data with patient history, thereby improving early lung cancer detection and prediction. Moreover, the research envisions expanding the model to handle multi modal data sources, including text, audio, and video, which could lead to a more comprehensive understanding and improved outcomes in lung cancer diagnostics. Future work should focus on improving image quality, expanding datasets, and addressing segmentation limitations to advance lung cancer detection.

### Author contributions

**Conceptualization:** Emad Shweikeh, Joan Lu, Murad Al Rajab.

**Data curation:** Emad Shweikeh, Abderahman Ahmed.

**Formal analysis:** Emad Shweikeh, Abderahman Ahmed.

**Investigation:** Emad Shweikeh.

**Methodology:** Emad Shweikeh, Joan Lu, Murad Al Rajab, Qiang Xu, Abderahman Ahmed, Hong Chang.

**Project administration:** Emad Shweikeh, Joan Lu.

**Resources:** Emad Shweikeh.

**Software:** Emad Shweikeh, Abderahman Ahmed.

**Supervision:** Joan Lu, Murad Al Rajab.

**Validation:** Emad Shweikeh, Joan Lu, Murad Al Rajab, Qiang Xu, Hong Chang, Mike Joy.

**Visualization:** Emad Shweikeh.

**Writing – original draft:** Emad Shweikeh.

**Writing – review & editing:** Emad Shweikeh, Joan Lu, Murad Al Rajab, Qiang Xu, Hong Chang, Mike Joy.

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
