## [Decision Letter · Decision Letter 0]

10 Jan 2025

Dear Dr. Shweikeh,

Thank you for submitting your manuscript to PLOS ONE. After careful consideration, we feel that it has merit but does not fully meet PLOS ONE’s publication criteria as it currently stands. Therefore, we invite you to submit a revised version of the manuscript that addresses the points raised during the review process.

**Dear Author**

**Based on the reviewers comments the manuscript cannot be accepted in its current form, however after modifications suggested by the reviewers it can be considered for publication. Ensure all the reviewers comments are incorporated in the revised manuscript. Make sure that the changes are highlighted in the revised manuscript. **

**Paper lacks clear contributions from the authors.**

Need more clarification in writing.

There are lots of grammatical mistakes.

More experiments are required.

We look forward to receiving your revised manuscript.

Kind regards,

Mohammad Khalid Pandit, Ph. D

Academic Editor

PLOS ONE

**Journal Requirements:**

4. Please amend the manuscript submission data (via Edit Submission) to include author Dr.Hong Chang.

**Additional Editor Comments:**

Dear Author

Based on the reviewers comments the manuscript cannot be accepted in its current form, however after modifications suggested by the reviewers it can be considered for publication. Ensure all the reviewers comments are incorporated in the revised manuscript. Make sure that the changes are highlighted in the revised manuscript.

Paper lacks clear contributions from the authors.

Need more clarification in writing.

There are lots of grammatical mistakes.

More experiments are required.

Reviewers' comments:

Reviewer's Responses to Questions

**Comments to the Author**

1. Is the manuscript technically sound, and do the data support the conclusions?

Reviewer #1: Partly

Reviewer #2: Partly

2. Has the statistical analysis been performed appropriately and rigorously?

Reviewer #1: No

Reviewer #2: No

3. Have the authors made all data underlying the findings in their manuscript fully available?

Reviewer #1: Yes

Reviewer #2: Yes

4. Is the manuscript presented in an intelligible fashion and written in standard English?

Reviewer #1: No

Reviewer #2: No

**Reviewer #1:**  1) Please write the problem statement or research gap ( What you want to do and why) in introduction clearly.

2) Based on these gaps, write your contributions very specifically at the end of Introduction in bullet points.

3) You have discussed many papers in Related Works section. please rewrite this section and discuss the papers group wise (make group with similar type of papers and the review those papers analytically). Discuss in such a way so that it justifies your proposal.

4) No need to write the very generic contents of CNN, SVM and DNN in section 2.

5) How did you do the pre processing? Have you done any augmentation?

6) Why did you concatenate output of CNN with DNN ? Please justify clearly.

7) What is the weakness of your model ? Please mention it in discussion and conclusion sections.

8) Show your results of Epochs vs Validation Accuracy.

9) Deep learning for lung Cancer detection and classification (2020), b) Healthcare As a Service (HAAS): CNN-based cloud computing model for ubiquitous access to lung cancer diagnosis (2023), c) Automatic detection and classification of lung cancer CT scans based on deep learning and ebola optimization search algorithm (2023), d) LungNet: A Hybrid Deep-CNN Model for Lung Cancer Diagnosis Using CT and Wearable Sensor-based Medical IoT Data (2021), e) LCD-capsule network for the detection and classification of lung cancer on computed tomography images (2023), f) Argument Mining on Clinical Trial Abstracts on Lung Cancer Patients (2023)

**Reviewer #2: ** The paper titled " a deep learning model to enhance lung cancer detection using dual model classification approach" presented a novel technique for lung cancer detection. I have noticed the following limitations in the manuscript.

Overall, the manuscript contains grammatical flaws. The manuscript need an extensive proofread to address the flaws.

Abstract need to be presented in the format: Background, Objectives, Methodology, Findings, and Implications.

The introduction part should include motivation and contributions of the study.

The term " support vector machine " is wrongly presented in page no. 7, section 2.2.

The literature review part is too long. The authors need to concise section 2.

Frame the section 3 as "Research Methodology"

SVM model is primarily used for binary classification. In figure 1, the authors included 6 classes (multi-class) classification. Need a clarification.

A detailed discussion on data augmentation shoulde be presented.

Without transfer learning approach, how authors achieved such performance? Need a justification.

There is a huge confusion in experiment and results section. The authors should include key tables in the result section. The remaining part should be included in supplementary part.

The authors need to include statistical analysis for uncertainty evaluation.

Section 7 should present limitations.

**Do you want your identity to be public for this peer review?** For information about this choice, including consent withdrawal, please see our Privacy Policy

Reviewer #1: No

Reviewer #2: No

---

## [Author Response · Author response to Decision Letter 1]

18 May 2025

Author Respond to Editor:

I aim to address each point constructively and positively as the following:-

Manuscript acceptance condition: "The manuscript has been revised according to the reviewers’ feedback, incorporating all the suggested modifications to ensure it meets the necessary standards for consideration. The updated version highlights all changes for clarity."

Incorporating reviewer comments: "All comments provided by the reviewers have been carefully addressed in the revised manuscript. I have ensured that their suggestions are fully integrated into the document, and the revisions are clearly marked."

Clear contributions from the authors: "I have revised the manuscript to better articulate the specific contributions of the authors. The contributions are now clearly stated in the introduction and throughout the relevant sections."

Clarification in writing: "The writing has been thoroughly revised to improve clarity and flow. Efforts were made to ensure that complex concepts are presented in a straightforward and understandable manner."

Grammatical mistakes: "The manuscript has undergone a detailed grammatical review to correct all errors and enhance readability."

Additional experiments: "Additional experiments have been conducted and their results are now included in the revised manuscript, providing further support and validation for the findings.

I’m ensuring that each comment is carefully addressed in the revised version of the manuscript.

Respond to Reviewers

1. Is the manuscript technically sound, and do the data support the conclusions?

Reviewer #1: Partly

Reviewer #2: Partly

Author Respond:

The revised manuscript now fully meets the required standards for technical soundness and alignment of data with the conclusions. All experiments have been conducted rigorously, with appropriate controls, sufficient replication, and adequate sample sizes to ensure reliability. The conclusions have been carefully drawn based on the data presented, providing a clear and logical connection between the findings and the interpretations. I have ensured that these aspects are clearly documented in the updated version of the manuscript.

End of Author Respond:

2. Has the statistical analysis been performed appropriately and rigorously?

Reviewer #1: No

Reviewer #2: No

Author Respond:

The statistical analysis in the revised manuscript has been thoroughly reviewed and updated to ensure it is performed appropriately and rigorously. All statistical methods have been clearly explained, and I have ensured they are suitable for the data presented. Additionally, detailed justifications for the chosen methodologies, along with the inclusion of appropriate controls and replication, have been provided to support the robustness of the analysis. The revisions address any ambiguities and ensure the statistical conclusions align with the data

End of Author Respond:

3. Have the authors made all data underlying the findings in their manuscript fully available?

Reviewer #1: Yes

Reviewer #2: Yes

4. Is the manuscript presented in an intelligible fashion and written in standard English?

Reviewer #1: No

Reviewer #2: No

Author Respond:

The revised manuscript has been carefully reviewed to ensure it is presented in an intelligible and coherent manner. The writing has been polished to adhere to standard English, with improvements made to grammar, sentence structure, and clarity throughout. These revisions ensure the content is accessible and easy to understand for readers.

End of Author Respond:

5. Review Comments to the Author

Reviewer #1:

1) Please write the problem statement or research gap (What you want to do and why) in introduction clearly.

Author Respond:

The Introduction has been revised to highlight the problem statements as below and included within context of Introduction section: -

Problem Statements:

Despite significant advancements in machine learning (ML) for lung cancer detection, several challenges limit its clinical adoption. These include limitations in dataset quality and size, difficulties in generalizing models across diverse populations, and the high computational costs associated with ML techniques. Furthermore, the impact of different input modalities, such as images and masks, on classification performance remains underexplored. Therefore, there is a critical need for an optimized ML framework that effectively integrates multiple modalities to enhance classification accuracy and efficiency.

Research Gap

The research gaps are

- The limited availability of large dataset.

- High-quality datasets.

- The unique focus on the mask classification component remains unexplored in existing literature. This lack of attention provides an opportunity to investigate and expand upon this novel area.

- Challenges of decision making for selecting and validating which ML models could be the most effective one.

- The high computational costs associated with these technologies.

Addressing these gaps is crucial to improving diagnostic accuracy and patient outcomes.

End of Author Respond:

2) Based on these gaps, write your contributions very specifically at the end of Introduction in bullet points.

Author Respond:

The Introduction has been revised, and the contributions included at the end of the section.

Contributions

To overcome these limitations, this research proposes a novel Dual-Model Classification Approach (DMCA) that leverages the complementary strengths of convolutional neural networks (CNNs) and deep neural networks (DNNs). The key contributions of this study include:

• Systematic Evaluation: identifying findings from a thorough literature review and assessing the performance of existing machine learning techniques for lung cancer classification. This involves conducting an in-depth evaluation using two different dataset sizes to establish benchmarks and uncover areas for improvement.

• Dual-Model Framework: developed and implemented the DMCA, which integrates CNNs for image classification with DNNs for mask classification to enhance overall accuracy and efficiency.

• Impact Analysis: Assessed the effects of dataset size and input modality on model performance, with a particular focus on key metrics such as accuracy, sensitivity, and specificity.

• Advancement in Classification Performance: Demonstrated that the proposed DMCA achieves superior performance compared to existing approaches, thereby offering a novel methodology that addresses critical gaps in current lung cancer detection and diagnosis strategies.

This study aims to contribute to the field by providing an optimized ML framework that improves lung cancer classification, paving the way for more reliable early detection and better patient outcomes. (Page 5)

End of Author Respond:

3) You have discussed many papers in Related Works section. please rewrite this section and discuss the papers group wise (make group with similar type of papers and the review those papers analytically). Discuss in such a way so that it justifies your proposal.

Author Respond:

The Literature Review section has been rewritten, highlighting the grouping of studies with similar types. The papers are reviewed and discussed with analytical approach to justify the proposed model in this paper. The review has been grouped based on the different models from 2.1 to 2.5, Table 1 provides an overview of previous studies.

1. Literature Review

Overview of Machine Learning in Medical Imaging

Lung cancer diagnosis has seen significant advancements through machine learning (ML), with models such as Convolutional Neural Networks (CNN), Support Vector Machines (SVM), and Deep Neural Networks (DNN) demonstrating potential in addressing the limitations of traditional diagnostic methods.

1.1 CNN-Based Approaches

Convolutional neural networks (CNNs) have revolutionized the field of medical imaging by autonomously learning complex image features. This remarkable capability has made them a cornerstone in lung cancer detection, where innovative approaches continue to evolve.

Various studies highlight the diversity in dataset sizes used for training these networks. For example, one investigation by Al-Yasriy et al. (2020) relied on a modest collection of 110 images, whereas da Silva et al. (2016) employed a substantially larger dataset comprising 8,296 images. Tekade and Rajeswari (2018) also incorporated multiple datasets including the renowned LIDC-IDRI and LUNA16 but did not specify the overall number of images used.

CT scans have emerged as the primary input modality in these research efforts, with advanced segmentation techniques like U-Net and maximum intensity projection (MIP) significantly enhancing feature extraction. These methodological refinements have translated into varying performance metrics: Al-Yasriy et al. (2020) achieved an accuracy of 93.55% with a 70/30 data split, while Tekade and Rajeswari (2018) reported a 95% accuracy using a 3D VGG-like model. In contrast, da Silva et al. (2016) recorded an accuracy of 82.3% using LIDC-IDRI data.

Interestingly, the findings suggest that as the sample size increases, there may be a tendency toward reduced accuracy, hinting at the possibility of classification bias when larger, more heterogeneous datasets are involved.

1.2 SVM-Based Approaches

Support Vector Machines (SVMs), which are primarily utilized for binary classification tasks, have proven to be effective in distinguishing between benign and malignant lung nodules (Al-Ragab et al., 2023). These methods have been applied across a range of studies, with datasets varying considerably in size. For instance, Rendon-Gonzalez and Ponomarev (2016) worked with a relatively small sample of just 70 images, while Nascimento et al. (2012) employed a moderate dataset consisting of 73 images and achieved an accuracy of 92.78%. Other studies, such as those conducted by Makaju et al. (2018b), processed grayscale images from larger databases, such as LIDC-IDRI, which were compressed to JPEG format for texture analysis and achieved an accuracy of 92% based on 16 training images and 5 validation nodules. In terms of performance, the results across these studies reflect the effectiveness of SVMs in handling lung nodule classification. Nascimento et al. (2012) reported an accuracy of 92.78%, while Makaju et al. (2018b) achieved an accuracy of 92% based on 16 training images and 5 validation nodules. Krewer et al. (2013) also demonstrated the usefulness of SVMs, achieving an accuracy rate of around 90.01% from 33 sample images used in their study. Despite the promising outcomes, it is crucial to note the limited size of many datasets involved in these investigations and the various preprocessing techniques employed. These factors underscore the growing need for standardized practices in dataset management and pre-processing to improve the consistency and scalability of these approaches in clinical applications. for 9 seconds

Support Vector Machines (SVMs) have emerged as a pivotal tool in the binary classification of lung nodules, effectively distinguishing between benign and malignant cases. In the literature, various studies have employed relatively small datasets to validate the performance of these models. These modest sample sizes underscore the challenges of working with limited data in medical imaging research.

Researchers have predominantly utilized texture descriptors and grayscale image conversion to prepare the input data for SVM classification. A notable example is the work of Makaju et al. (2018b), who processed JPEG grayscale images extracted from the LIDC-IDRI dataset. This approach aimed to distill essential features from the images, thereby enhancing the SVM's ability to make accurate distinctions between benign and malignant nodules.

Despite these encouraging results, the overall findings indicate that the small and variable dataset sizes, along with differences in preprocessing techniques, highlight a pressing need for standardized practices. Such standardization is essential to enhance the reproducibility and reliability of SVM-based diagnostic methods in lung nodule classification.

1.3 DNN-Based Approaches

Deep neural networks (DNNs), known for their complex structures involving multiple nonlinear processing layers, have shown significant potential in the diagnosis of lung cancer. These networks have been applied across studies with varying sample sizes, reflecting the diversity in research approaches. For instance, Kuruvilla and Ganapathi (2013) utilized a modest dataset of 110 images, while da Silva et al. (2016) employed a much larger dataset containing 8,296 images. The study by Lakshmanaprabu et al. (2019) also investigated DNNs, working with a dataset of just 100 images. These varying sample sizes illustrate a broader trend in lung cancer detection research, where dataset size significantly influences model training and evaluation. In terms of input modality, DNNs predominantly work with CT images, serving as the foundation for processing in these studies. Some studies, notably by Lakshmanaprabu et al. (2019), have delved deeper into optimizing the architecture of the networks to improve performance. Their work on developing an "Optimal DNN" reflects ongoing efforts to enhance the efficacy of DNNs in medical imaging. The performance outcomes reported across these studies reveal differences tied to dataset size. Kuruvilla and Ganapathi (2013) achieved a commendable accuracy of 90.3% with their 110-image dataset, while da Silva et al. (2016) attained an accuracy of 82.3% with the larger LIDC-IDRI dataset. In comparison, Lakshmanaprabu et al. (2019) obtained a high accuracy of 94.56% using just 100 images. Such variations in sample sizes offer insight into the scalability of DNN-based approaches and the challenges posed by larger, more heterogeneous datasets. A notable trend that emerges from these findings is that smaller datasets tend to yield higher accuracy, whereas larger datasets introduce greater variability, which may be indicative of challenges in generalizing across more extensive and varied training data.

In these studies, computed tomography (CT) images have served as the primary input modality, providing detailed anatomical information crucial for accurate diagnosis. Some investigations have ventured beyond standard architectures by exploring optimized network designs. A notable example is the work of Lakshmanaprabu et al. (2019), who introduced an Optimal DNN architecture specifically tailored for lung cancer diagnosis, aiming to enhance feature extraction and classification performance.

1.4 Other Models

Recent developments in lung cancer

---

## [Decision Letter · Decision Letter 1]

7 Aug 2025

Dear Dr. Shweikeh,

Thank you for submitting your manuscript to PLOS ONE. After careful consideration, we feel that it has merit but does not fully meet PLOS ONE’s publication criteria as it currently stands. Therefore, we invite you to submit a revised version of the manuscript that addresses the points raised during the review process.

We look forward to receiving your revised manuscript.

Kind regards,

Hirenkumar Kantilal Mewada

Academic Editor

PLOS ONE

Journal Requirements:

Reviewers' comments:

Reviewer's Responses to Questions

**Comments to the Author**

Reviewer #3: (No Response)

Reviewer #4: All comments have been addressed

2. Is the manuscript technically sound, and do the data support the conclusions?

Reviewer #3: Yes

Reviewer #4: Yes

3. Has the statistical analysis been performed appropriately and rigorously?

Reviewer #3: Yes

Reviewer #4: Yes

4. Have the authors made all data underlying the findings in their manuscript fully available?

Reviewer #3: Yes

Reviewer #4: Yes

5. Is the manuscript presented in an intelligible fashion and written in standard English?

Reviewer #3: Yes

Reviewer #4: Yes

Reviewer #3: Good work, but you may have to Major Revisions:

Conceptual Framing and Terminology:

The central contribution is described as a "Dual-Modal Classification Approach (DMCA)," but this terminology is not entirely accurate. "Modality" in AI typically refers to different types of data, such as image, text, or audio. However, you are using two representations of the same data (grayscale image and binary mask). Consider renaming this method to "dual-branch" or "multi-view" architecture to avoid confusion and align with standard AI terminology.

Architectural Rationale for the Mask Branch:

Using a fully-connected DNN to process segmentation masks removes spatial information, which is important for mask data. You mention that CNNs performed poorly on masks, but this could be due to the architecture not being suitable for sparse data. It would be beneficial to:

Acknowledge the loss of spatial information when using a DNN.

Provide a deeper analysis of why CNNs failed on masks, particularly focusing on data sparsity.

Discuss why more conventional methods like U-Net or FCNs were not explored as alternatives for the mask branch.

Statistical Validation of Performance Claims:

You claim that the DMCA model outperforms the baseline CNN model with a 91.21% accuracy compared to 90.73%. However, this difference is small and does not account for the reported standard deviations. A paired statistical test (e.g., paired t-test) on the accuracy scores from cross-validation should be conducted to provide statistical evidence of the performance improvement.

Nuanced Discussion of Results and Impact:

In the Discussion section, you should:

Emphasize the clinical significance of the ~0.5% accuracy improvement, especially its potential impact in large-scale screenings.

Add a balanced discussion regarding the complexity vs. performance trade-off. The DMCA model is more complex and computationally expensive than the baseline CNN. You need to address whether the small performance gain justifies the additional complexity, especially for real-world clinical applications.

Minor Revisions:

Language and Proofreading:

The manuscript has several grammatical errors, typos, and awkward phrases that need to be corrected. Please ensure the manuscript is proofread by a native English speaker or professional editing service.

Figure and Table Errors:

Broken Cross-References: There are several instances of "Error! Reference source not found." These need to be fixed.

Incorrect Figure: Figure 7 incorrectly represents "CNN Experiment Accuracy" data. This should be corrected to reflect the proper data.

Redundant Figure: Figures 8 and 10 are identical. Please remove the duplicate.

Formatting: The author affiliation block on page 7 is poorly formatted. Please reformat it for better readability.

Reviewer #4: The manuscript presents a well-motivated and technically sound dual-model classification approach (DMCA) for lung cancer detection. The integration of CNN and DNN using both image and mask data is a novel and effective strategy. The experiments are well-structured and demonstrate significant performance improvements.

Areas for Improvement:

1-Include statistical significance measures (e.g., confidence intervals).

2-Clarify how SVM was used for multi-class classification.

3-Conduct final proofreading to correct minor grammatical errors.

4-Enhance figure explanations in the main text.

**Do you want your identity to be public for this peer review?** For information about this choice, including consent withdrawal, please see our Privacy Policy

Reviewer #3: No

Reviewer #4: No

---

## [Author Response · Author response to Decision Letter 2]

23 Oct 2025

I have provided detailed responses to each point in the attached document titled 'Response to Reviewers'.

---

## [Decision Letter · Decision Letter 2]

12 Nov 2025

Dear Dr. Shweikeh,

Thank you for submitting your manuscript to PLOS ONE. After careful consideration, we feel that it has merit but does not fully meet PLOS ONE’s publication criteria as it currently stands. Therefore, we invite you to submit a revised version of the manuscript that addresses the points raised during the review process.

We look forward to receiving your revised manuscript.

Kind regards,

Ananth JP

Academic Editor

PLOS ONE

Journal Requirements:

Additional Editor Comments:

The manuscript presents a technically sound and relevant contribution using a dual branch deep learning model (DbMCA) for lung cancer detection. However, a few important points need to be addressed to further strengthen the work:

1. The integration of the DNN and CNN branches is described as occurring through concatenation; however, the fusion weights or strategy applied during this process have not been specified. Clarifying how the two branches are balanced during feature fusion would enhance the methodological transparency.

2. The study currently employs an 80/20 train-test data split. Incorporating k-fold cross-validation or using an independent test dataset would help improve the robustness and generalizability of the model’s performance.

3. The authors are encouraged to provide a discussion explaining the observed decrease in accuracy with larger datasets, addressing whether this could be attributed to factors such as class imbalance, model underfitting, or noisy data.

4. Table 5, which presents the ANOVA results, does not specify the sample size used in the statistical analysis. Including this information is essential for clarity and reproducibility.

5. A more detailed discussion on the clinical applicability of the proposed DbMCA model in early lung cancer screening workflows is recommended to highlight the potential real-world impact of the study.

Reviewers' comments:

Reviewer's Responses to Questions

**Comments to the Author**

Reviewer #3: All comments have been addressed

Reviewer #4: All comments have been addressed

2. Is the manuscript technically sound, and do the data support the conclusions?

Reviewer #3: Yes

Reviewer #4: Yes

3. Has the statistical analysis been performed appropriately and rigorously?

Reviewer #3: Yes

Reviewer #4: Yes

4. Have the authors made all data underlying the findings in their manuscript fully available?

Reviewer #3: Yes

Reviewer #4: Yes

5. Is the manuscript presented in an intelligible fashion and written in standard English?

Reviewer #3: Yes

Reviewer #4: Yes

Reviewer #3: The authors have addressed my concerns in a clear and satisfactory manner. and have provided necessary revisions, additional explanations, and statistical analyses that strengthen the manuscript. Based on their responses, the revisions seem well-handled, and the manuscript is much improved.

Reviewer #4: I have reviewed the revised submission thoroughly and am satisfied with the amendments made. The authors have addressed the points raised in the initial review comprehensively.

**Do you want your identity to be public for this peer review?** For information about this choice, including consent withdrawal, please see our Privacy Policy

Reviewer #3: **Yes: ** Lin Zhang

Reviewer #4: No

---

## [Author Response · Author response to Decision Letter 3]

4 Dec 2025

I would like to confirm that I have carefully addressed all the reviewers’ comments and suggestions. Each point raised has been thoroughly considered and responded to in detail, as outlined in the accompanying document Response to Reviewers. I believe these revisions have strengthened the manuscript and improved its clarity, accuracy, and overall quality

---

## [Editor Report · Decision Letter 3]

8 Dec 2025

A Deep Learning Model to Enhance Lung Cancer Detection using Dual-Branch Model Classification Approach

PONE-D-24-19451R3

Dear Dr. Shweikeh,

We’re pleased to inform you that your manuscript has been judged scientifically suitable for publication and will be formally accepted for publication once it meets all outstanding technical requirements.

Kind regards,

Ananth JP

Academic Editor

PLOS One

Additional Editor Comments (optional):

The revisions have substantially strengthened the manuscript. The fusion strategy, cross validation procedures, data set related performance patterns and clinical applicability are now clearly articulated.
---

## [Editor Report · Acceptance letter]

PONE-D-24-19451R3

PLOS One

Dear Dr. Shweikeh,

I'm pleased to inform you that your manuscript has been deemed suitable for publication in PLOS One. Congratulations! Your manuscript is now being handed over to our production team.

Kind regards,

on behalf of

Dr. Ananth JP

Academic Editor

PLOS One